# Doing Everything We Can to Help Our High-Risk Newborns: A Qualitative, Lifeworld-Led Study of What Early Risk Assessment for Cerebral Palsy Means to Parents

**DOI:** 10.3390/jcm14082740

**Published:** 2025-04-16

**Authors:** Kristin Bjørnstad Åberg, Karin Dahlberg, Gunfrid Vinje Størvold, Ragnhild Støen, Lars Adde

**Affiliations:** 1Department of Clinical and Molecular Medicine, Norwegian University of Science and Technology, 7491 Trondheim, Norway; ragnhild.stoen@ntnu.no (R.S.); lars.adde@ntnu.no (L.A.); 2Department of Pediatrics and Habilitation, Levanger Hospital, Levanger, Nord-Trøndelag Hospital Trust, 7601 Levanger, Norway; gunfrid.v.storvold@ntnu.no; 3Independent Researcher, 51891 Sjömarken, Sweden; karin.dahlberg@pajebogard.se; 4Regional Centre for Habilitation, Department of Mental Health, Norwegian University of Science and Technology, 7491 Trondheim, Norway; 5Department of Pediatrics, St. Olavs Hospital, Trondheim University Hospital, 7491 Trondheim, Norway; 6Clinic of Rehabilitation, St. Olavs Hospital, Trondheim University Hospital, 7491 Trondheim, Norway

**Keywords:** cerebral palsy, early predictive assessments, general movement assessment, parental perspectives, qualitative research, phenomenology

## Abstract

**Background/Objectives:** Early predictive assessments for CP are recommended for infants with medical risk factors after birth. For parents of children with CP, receiving an early diagnosis is important. But most children with risk factors who have not yet developed CP are labeled “high-risk infants” and repeatedly assessed for abnormal signs. We aim to investigate the experience of parents of high-risk infants and describe the meaning that “early predictive assessments for CP” has for them before they know whether their children have CP. **Methods:** This was a qualitative study conducted using a phenomenological, reflective lifeworld approach. Fourteen individual in-depth interviews were conducted with parents who received different GMA results to learn about their experiences involving early predictive assessments. The interviews were analyzed for meaning. **Results:** Early predictive assessments take place over time while parents process the traumatic experience of becoming parents to an infant at risk. “Early predictive assessment” is perceived as any examination or assessment intended to unveil signs of illness or disability. The child’s future well-being and fulfillment, and the demands of parenthood, are at stake. Essential meaning structures are (1) on a spectrum from death to insignificancies, (2) living with uncertainty of what the parental role will entail, and (3) seeing one’s own child through the eyes of strangers, just in case. **Conclusions:** For months following the birth of a high-risk child, parents experience uncertainty and worrying, affecting the parent–infant relationship. Predictive assessments reduce their sense of alarm when the GMA result indicates a low risk of CP. But when the GMA result is uncertain, the burden of uncertainty is amplified and prolonged.

## 1. Introduction

Brain injury after perinatal asphyxia, stroke, cerebral infection, or prematurity is associated with an increased risk of cerebral palsy (CP) and other adverse neurodevelopmental outcomes [1,2,3]. International and Norwegian clinical guidelines recommend that infants with known medical risk factors, so-called high-risk infants, are followed up by multidisciplinary teams and assessed with appropriate tools for the early detection of CP [4,5,6]. Early assessment enables an early interim clinical diagnosis of high risk of CP [5,7]. The goal is to facilitate early, diagnosis-specific interventions aimed at reducing the severity of symptoms, optimizing future function, preventing complications, and offering parental support [4,7,8].

The General Movement Assessment (GMA) is the most accurate predictive assessment tool for determining CP before 5 months age [4,5,6,9]. The GMA is a highly specialized method to assess the quality of infants’ spontaneous movements, so-called fidgety movements, observed in video recordings [10,11]. It is currently not universally available in follow-up programs for high-risk infants, but accessibility is expanding as infrastructure is being developed for remote GMA using digital platforms [12,13,14,15,16,17].

Besides identifying infants with a high risk of CP, identifying infants who are unlikely to develop CP is considered beneficial for parents of infants with recognized risk factors after birth. Such early reassurance supports medical professionals in decision-making regarding follow-up needs and helps them prioritize limited health care resources. In recent years, several studies have investigated parents of children with CP for their needs regarding early diagnosis and early intervention programs. Overall, these studies found a favorable attitude towards early assessments and early, targeted intervention among parents of infants who develop CP [18,19,20,21,22].

However, most infants with recognized medical risk factors do not develop CP. The prevalence of CP ranges from around 30% after stroke to around 7% for very preterm, and it increases with decreasing gestational age [3,23]. Studies targeting parental experiences with early prediction rarely include the perspectives of parents to children who turn out not to have CP despite medical risk factors being identified during the peri- and neonatal period. These children are nonetheless labeled as high-risk infants from the onset of their lives and are included in extensive follow-up programs due to the increased risk of adverse neurodevelopmental outcomes [24,25]. The impact on parents who are exposed to early predictive assessments of their child is not well described. Furthermore, early predictive examinations will never be perfect [9]. False positive results or uncertain results have the potential to worry parents, and the term high-risk may be emotionally loaded for parents when applied to their newborn child.

In this study, we aim to investigate the phenomenon of early predictive assessment for CP as it is experienced by parents of infants included in a high-risk follow-up program after discharge from neonatal care. We aim to describe the essential structures of meaning that this phenomenon holds for parents at a point in time when definitive neurological signs of CP are not yet apparent, and a CP diagnosis is not yet made nor excluded. For the purpose of exploring the phenomenon in full, with variations and nuances that may appear in different parental experiences, we want to include different GMA results, i.e., from parents who were informed that (1) their child had a high risk of CP (fidgety −), (2) their child had a low risk of CP (fidgety + and fidgety ++), and (3) their child likely did not have CP but the examination was not entirely normal (fidgety +/−) or the results varied between different videos. For this purpose, our research question is “What is the meaning of early predictive assessment for CP for parents of infants with known medical risk factors after birth?”

## 2. Materials and Methods

This study is reported according to the Standards for Reporting Qualitative Research (SRQR) [26], Appendix B.

Phenomenological methods are well suited when attempting to understand the meaning of people’s experiences with health care [27]. A reflective lifeworld research approach (RLR) and a specific procedure for data collection were chosen and adapted through close discussions with experts in qualitative methodology. The epistemological background of RLR stems back to Husserl’s and Merleau-Ponty’s phenomenology with a focus on lifeworld theory, aiming to identify and describe essential meaning structures of a phenomenon as it appears to the persons experiencing it [28,29]. RLR, when applied in medicine and health research, sees health care as a meaningful practice where human experiences unfold as narratives best understood through “meaning” rather than measurement [30]. Epistemologically, RLR represents a sensible “middle ground” alternative to the phenomenological traditions of interpretative phenomenological analysis (IPA) and descriptive phenomenological analysis [29]. Methodological principles central to RLR are being phenomenon-oriented and lifeworld-led and practicing openness, bridling, and reflectivity. RLR acknowledges researchers´ pre-understanding but emphasizes their ability to bridle, i.e., slow down the process of understanding, to not take emerging results for granted but challenge them.

Parents of infants who were referred to neurodevelopmental follow-up after neonatal care were recruited from three hospitals in the Central Norway Regional Health Authority during the 2022–2023 period. Informants were selected among participants in a parallel, ongoing multicenter study (in-motion study) on the feasibility of home video recordings by parents using their own cell phones as a basis for remote general movement assessment (GMA) [14]. All informants had experience with remote GMA based on two or three home videos and one video recorded by health care personnel (HCP) in their local hospital. Parents had received the result of the GMA analyses either via letter, phone call, or an in-person meeting with a contact person from their follow-up team. They had not yet had a scheduled clinical follow-up with a pediatrician to assess for definitive neurological signs of CP. The results of the GMAs were presented as a low risk of CP for nine parents and uncertain for three parents. Two parents had received contradicting results from different GMA. They were first informed that there was a high risk of CP in three videos; weeks later, they were told a fourth video showed a low risk of CP.

Inclusion criteria were parents of infants participating in a parallel, ongoing study (in-motion study) [14] in which a remote GMA was performed, who received the results of the GMA and were able to communicate fluently in Norwegian. Exclusion criteria were those who were informed about a certain CP diagnosis due to definitive signs on neurological examination, foster care placement, and death of the child. We included participants using a strategic sampling strategy. Strategic selection criteria and sample characteristics are listed in Table 1. Participants were chosen in collaboration with health care personnel from follow-up programs in the collaborating hospitals, aiming for diversity in gender, age, family structure, health status, education, rural versus urban residence, and cultural backgrounds. Selected participants were then approached by the project leader of the in-motion study [14], asking permission for the first author to contact them with information about this study and an invitation for interview. Fifteen parents were approached, and two declined participation. One that was not approached initiated contact with the first author expressing an interest in being interviewed, making the total number of interviews 14.

Data were collected between May 2022 and September 2023; 14 individual in-depth interviews were conducted by the first author either in the informants’ homes, a meeting room in the nearest hospital, or via videocall. Four initial interviews were guided by a semi-structured interview guide which, after a preliminary analysis, was found to restrict informants’ ability to elaborate on topics important to them. The guided interview process was then replaced with an open interview strategy designed to set the stage for an atmosphere of mutual trust and confidentiality and to allow the informants to set the agenda in an open conversation centering around their experiences with early predictive examinations, with the interviewer asking for elaborations and specific examples to elicit pre-reflexive lived experience descriptions. Initial interview guide (Appendix A) and revised interview strategy (Appendix A) are available as Appendix A. Interviews were audio-recorded and transcribed verbatim by the first author. Transcripts were de-identified, and transcripts and audio-recordings were stored in an access-controlled, secure server at St. Olavs Hospital. After conducting 13 interviews, we considered the data to be of sufficient richness and depth and that more interviews would likely not add more significant meaning. However, the last one was already scheduled and was therefore also included.

For the sake of reflexivity, we disclose that the first author is a pediatrician with experience in pediatric neurology and habilitation, including follow-up of high-risk infants. She has not had any prior contact with any infants or parents involved in this study, but informants received information about her background as a pediatrician during recruitment. For the sake of reflexivity and transparency, prior to drafting an interview guide, she wrote down her pre-conceived notions about parents’ experiences and preferences. She believed that parents mostly have a favorable attitude toward early prediction. Dwelling on her own experiences as a pediatrician in neonatal follow-up and multidisciplinary habilitation teams, and how this experience from a professional’s perspective does not equal true insight in the first-hand experience of parents, she attempted to approach every interview and the analysis with mindful openness, bridling impulses, and suspended judgment so that unexpected facets and meanings could emerge.

During analysis, all transcripts were read multiple times to acquire familiarity with the material as a whole and each informants’ unique point of view. The first author read all interviews, while the second, third, and last authors read the most information-rich interviews. The first author then turned to the particulars, re-reading the transcripts with a focus on meaning units, naming preliminary topics and organizing statements from all interviews. Microsoft Word was used for transcribing and organizing data during analysis with the addition of hand-drawn mind maps. The preliminary topics were discussed with the second author in several phases. Through an iterative process, going back and forth between the details and the whole material in the interviews, the topics were revised and redefined into three tentative essential meaning structures. When writing the manuscript, ChatGPT (OpenAI, GPT-3.5 model) was used for help with the optimal translation of some individual words and short phrases from Norwegian to the English language in Section 3 and in some of the participant quotes. To ground the results in the narratives of the informants, and to fully understand and be able to describe the depth and variability of the meaning structures, for each informant, the following questions were asked: (1) “Is this meaning present in this interview? (2) “Is this a good way of understanding this person’s experience?” The first part of the results defines the phenomenon as it appears in this study and its relation to the broader context of parents’ lived experiences (i.e., the phenomenon’s outer horizons) [31]. Three essential meaning structures follow. They comprise both essential and more general meanings and have individual as well as contextual constituents. Quotes from the interviews illustrate these meanings.

## 3. Results

### 3.1. Essential Structures of Meaning

Parents’ vantage point of early predictive assessments for CP dictates how the phenomenon is described in this study. The phenomenon of early predictive assessments appears to parents while they experience becoming parents to a sick, injured, or premature child where there is a risk for lasting neurological impairment. This risk gives the indication for early predictive assessments, and early predictions begin as soon as the risk is acknowledged. Predictive assessments are broadly perceived as any assessment or examination which may unveil signs of future disability or illness. They are performed repeatedly in the weeks and months following the child’s birth by many different HCPs. Parents live, and cope, with the risk first in the hospital and then in their homes while repeated predictive assessments occur.

The essential meaning of early prediction for CP as experienced by the parents is generated in the interplay between the trauma of having a newborn at risk and new parenthood. It is characterized by a desire to know that the child will live and be well. The context of what these parents are living through as they participate in the assessments cannot be separated from or omitted when attempting to understand the significance of the patients’ experiences of early prediction. Their perspective is framed by trauma and insecurity. The risk posed to their child is about more than CP. It encompasses any disability or illness which may threaten the well-being of their child and their future life. Their experiences of risk together with predictive assessments target the existential question of what it will mean to fulfill the parental role for their child.

The meaning structure of how early predictive assessment for CP as experienced by parents can be further displayed by three themes of meaning: 1: on a spectrum from death to insignificancies, 2: living with uncertainty of what the parental role will entail, and 3: seeing one’s child through the eyes of professionals just in case.

#### 3.1.1. On a Spectrum from Death to Insignificancies: If My Child Survives This, What Sort of Life Will They Have?

Early predictive assessments begin the moment parents become aware that there is a risk that their child may suffer long-term consequences from their neonatal illness. This may begin gradually, several weeks before preterm delivery, or suddenly and unexpectedly when a lifeless newborn is delivered after shoulder dystocia. In either scenario, there may be a momentary realization that their child may not live. That is a threat to the existence of their parenthood, to their being as parents. A mother recalls her thoughts in this moment: “*Okay, he has been without oxygen a long time now. Will he survive? (…) After a while I got him on my chest with [respiratory support]. (…) Then I was able to abandon the thought of him not being alive anymore. But I remember my first thought as a mother being—If he dies now, I do not want to live anymore*”.

When parents realize that their child will live, predictive assessments turn from live or die to whether the child will experience much suffering and what sort of life the child will have. From now on, assessments are experienced from the following point of view: it could have been worse; the child could have been dead. Before any information is shared and before the result of any examinations are given by an HCP, parents make their own observations and assumptions about the severity of what has happened and to what extent the future of the child is threatened. A father tells us the following: *“Both the mother and I thought we would lose him when he was born. Then we thought he would become a vegetable. But as we rolled him over to the NICU, he tried to pull out his own tube. When we got there, he was on my chest without any respiratory support, and it all looked very good. So, then we got the faith back*”.

While the worst-case scenario quickly fades to something less ominous for some, others continue to experience this life-or-death perspective for a long time. If the child’s risk of dying at one time was very high or is perceived to be recurring, for instance, a with recurring respiratory problems, parents express an emphasis on being present in the moment and enjoying time with their child rather than worrying about the future. The possibility of one’s child suddenly dying eclipses any notion of hypothetical future disabilities, making predictive assessments for CP comparatively less important. Others have a range of concerns for the future. Any condition which may threaten the child’s future well-being, fulfillment, and ability to embrace life is weighed, graded, and placed along a figurative spectrum of severity. The spectrum is not reserved for individual constructs such as CP. Vision and hearing impairment, ADHD, cognitive impairment, epilepsy, heart disease, pulmonary disease, and psychological issues, all of which parents know that high-risk children are at risk of, are included. Thus, although GMA is specifically used for risk assessments for CP, it still falls in the same category of examinations as screening for hearing impairment, screening for retinopathy of prematurity, or pulmonary function tests, as this mother shows us: *“It’s the same as our check-up at the pulmonologist. If she needs a nebulizer she needs a nebulizer. Then we just have to deal with that”.* What constitutes a predictive assessment for parents is something which informs them where on the spectrum of severity to ground their worries.

As weeks and months pass, parents perceive predictive assessments as repeated examinations, opinions, and advice from NICU nurses, physiotherapists, pediatricians, primary care nurses, family doctors, and even from family and friends. The child’s movements, positioning, and skills are repeatedly scrutinized, deciphered, and translated into information on how their child will likely fair. Early predictions are ongoing throughout infancy and occur within the relationships that develop between parents and HCPs. Their results are seen in relation to other factors which also influence parents’ perceptions of their situation. One mother talks of how she takes comfort in being reminded that things could have been worse: “*I saw a mother with a child which was severely disabled. Then I was able to see that perspective too, and to think that it could have been so much worse. [My child] could have not survived, for that matter. I try to hold on to that thought and find courage in it*”. A similar sentiment of perceiving their situation relative to what others go through is expressed by another parent: “*When we lived [in the NICU] and heard others’ stories, we felt like we weren’t allowed to complain, really*”. Thus, the gravity of parents’ concerns is informed by the results of early predictive assessments, yet it also depends on contextual factors. What their child’s “worst-case scenario” was, whether they know other children with disabilities, and parents’ own health and financial situations influence the magnitude of parents’ concern for their children’s future and their placement along the spectrum of severity.

For some parents, a GMA result indicating a low risk of CP leaves nothing but “insignificancies” to worry about. As they observe their child thriving and one professional after the other confirms good development, the GMA result joins in the string of assessments that help reduce their sense of alarm. “*It means we can lower our shoulders, look ahead, plan a little more*”, a mother explains. A GMA result indicating a high risk of CP, while causing sorrow and pain, may not be surprising to parents. Their own gut feeling and several conversations with HCPs over time have prepared them for this possibility. Two notions modify the reception of the bad news: “*At least my child is alive*” and “*CP can be many things; it can be mild*”. Hence, the message of a high risk of CP is not enough to inform parents about the real-life implications for their child’s future.

The GMA result of fidgety +/− and opposing results after different videos are perceived by some parents as uncertain. They then continue to be unable to decisively place their concern on the figurative spectrum of severity. These results indicate to parents that something may not be right, though it is likely not CP. The threat to their child’s future becomes more real and tangible, yet still undefinable and elusive. What follows is amplified uncertainty which lasts longer than they had hoped.

#### 3.1.2. Living with the Uncertainty of What the Parental Role Will Entail: If My Child Will Be OK, I Will Be OK

Uncertainties for a child’s prospects impact how parents envision their lives with the child and what the parental role will entail. Parenting a child with CP or other chronic illnesses has practical implications for how a family can live their lives, and some parents start to prepare practically and mentally for a reorientation of their lifestyle and future plans. “*For instance, if she’ll need an assistive device, like a walker or something. How will we manage that, when we have stairs? How are we going to manage…? I mean, things that don’t really need to be a problem, but you end up thinking about them. How are we going to go on mountain hikes? Things like that. You kind of dwell on thoughts like that*”.

Beyond such practicalities, we find a strong emotional connection between parents and their child, where the child’s future well-being strongly impacts the parent’s own sense of well-being in the moment. The notion that an infant is entirely dependent on caregivers to thrive and that the nature of parenthood includes living for someone else is very notable for many parents in this study. “*I can’t be so selfish that I make decisions without thinking about the child first. (…) She gets all the attention and strength I have left*”, said a mother explaining how a major life decision had been impacted by the possibility that her sick child might have extraordinary needs in the future. This connection is also evident for parents who struggle with health-related uncertainties for their own future. In one parent’s words: “*I may not be able to help her as much as I want to. (…) I used to think a lot about what if we both become wheelchair bound? That’d be a heavy load for my family. (…) but now that I see how she is doing, I’m thinking it’ll be all right. (…) That thought lifts me up. When I’m thinking she’ll be fine, I’m fine*”.

Parents who focus on the risk highlight uncertainties not only for the child’s prospects for health, happiness, and fulfillment, but also for the parents’ own lives and what it takes to be good parents for a child with special needs. The uncertainties are part of their life for weeks and months after leaving the hospital and become part of their growing relationship with a brand-new human. Some parents speak of limiting how much room worrying about the future gets in the way of their everyday lives, while others find it hard to obtain control over the worrying. A father tells us how he often thinks about what the future may demand of him if his child has a disability, but he keeps the pondering at bay: “*I may watch her sleeping in her bed, and start thinking “if this happens, we may have to do that (…) But it’s not something I think about all the time. Because if I would have done that… It’s exhausting, dwelling on the “what-if’s” all the time*”. Meanwhile, a mother describes how worrying significantly affected her days and her connection with the child. In her words, “*I have focused on following his development closely, looking for signs. I have read so much about CP, I’ve joined CP-groups on Facebook. I’ve gone all in preparing, in a way. And I think it’s made me unable to see all that has been positive. I haven’t been able to enjoy the first year with my kid. It’s brutal to say it, but I haven’t been able to relax. And I think ahead a lot. If he gets CP, what sort of life will he get?*” Both statements indicate how worrying about the future impacts daily life and is emotionally draining. When dealt with and contained, worrying can help parents prepare practically, economically, and psychologically for a future which may be demanding for the whole family. Knowing that they are as prepared as possible for any eventualities helps parents relax and enables them to tune in to the present demands of the child and everyday hustles, and the risk may fade into the background. But some have a high level of anxiety and that is a thief of time, mental presence, and energy. Worrying is exhausting. For some parents, tuning into their child and being present in the moment while preparing for possibly debilitating illness in the future are contradictory exercises, and they struggle with integrating these seemingly opposite demands on their parental duties.

Furthermore, a pair of parents may not share the same attitude towards balancing worrying and preparing for the worst. Within a relationship, partners can find validation and comfort, while also being challenged on their intuitive reactions. Some parents speak of spending much time in conversation about how to relate to risk and prepare for the future, and some express a sense of being grounded and brought back to the present by a level-headed partner when succumbing to catastrophic thoughts. At the same time, not sharing a similar intuitive sense of alarm may cause loneliness within a partnership.

This loneliness is exacerbated by the experience that family and friends lack an understanding of what it is like to care for a high-risk infant, what they have lived through, and the tolls of living with uncertainty. Integral to the concept of early predictive assessments for CP is the fact that the potential illness is invisible to the layman’s eye, and it follows that the child’s needs and situation may not actually be how it appears. Here grows a sense that this child is not like other infants, and that a friend, uncle, or grandmother may not have the insights needed to understand this child’s needs. Some express a perceived lack of appreciation of this by friends and family who do not share their experience of parenting high-risk infants. Contrarily, sharing experiences and stories with other experienced parents of high-risk children during hospitalization, in patient support organizations, and on social media networks gives parents validation that living with uncertainty is difficult and reassures them of the normality of their feelings and reactions.

The burden of living with uncertainty for one’s child’s future is gradually lifted by the passing of time as parents begin to know their child, see their development, and see the presence or absence of clear signs of illness. It is also lifted by the significant amounts of feedback from predictive assessments made over time by nurses, physiotherapists, and doctors. It is eventually further lifted by the result of the GMA.

When predictive assessments indicate a low risk of CP, parents feel joy and relief. Still, some degree of uncertainty seems to remain with every version of result after the GMA. Even after a low-risk assessment, parents are aware of the possibility of a later appearance of ADHD and social or cognitive developmental delays. Parents in our study who received a GMA result of a high risk of CP expressed an understanding that the implication for the child and demand on the parents are incomparably different depending on the type and severity of CP. This shows that early predictive examinations for CP can never give a complete answer to the question of what sort of future a child faces.

And when the result of the GMA is inconclusive, the burden of living with uncertainty is amplified, as illustrated by this parent’s recollection of receiving the GMA result: “*I got scared. (…) Because even today we don’t know anything, and many things are not like normal. My child is behind in motor skills. And the GMA result didn’t tell me anything. (…) I’ve googled it, and understand it is 80–90% certain it won’t be CP. (…) But still it isn’t the normal result*”. Thus, some GMA results may confirm to parents that something is not right without giving any indication as to what this may be nor what it may imply for the child’s future. What is at stake for parents is the demands on their roles as safe guardians of their children’s future.

#### 3.1.3. Seeing Your Child Through the Eyes of Professionals, Just in Case: Help Me Give My Child What They Need

An explicit goal for parents in this study is to be able to set themselves up for whatever the future may throw at their child. The surface value of early predictive examinations serves this purpose for them. When they obtain an indication of whether their child has any illness or impairment, they can prepare practically, economically, mentally, and emotionally. They also appreciate the opportunity to be in contact with a follow-up team which has known their child from a young age and is prepared to guide the parents and help the child as early as possible.

The flip side of this is that parents must accept an extended outsider presence in their lives. This comes on top of the absence of normality they have already lived through at the onset of their parenthood, where the desire for a private family sphere is in stark contrast to the realities described by our informants. Parents to severely ill or premature infants gain familiarity with their new children under the watchful eyes of machines, nurses, doctors, and physiotherapists who monitor the children for abnormal signs. Parents learn to keep track of and interpret the numbers and signs. As they leave the hospital, some parents speak of intense fear as they lose the sense of control that monitoring gives, as well as the support and security from the proximity of nurses and doctors. They have gotten used to receiving continuous external confirmation that their child is well and stable. Shedding the wires, tubes, and machines that go “bing” in and of themselves makes the infant appear healthier, and coming home reintroduces normality in their lives. Yet for some parents, having to trust their own judgment on their child’s well-being also feels very insecure. The looming threat of a temporarily invisible illness, like CP, seizures, or even sudden respiratory arrest, lingers.

Thus, parents of high-risk infants have not gotten to know their child as “presumably healthy”. Illness or the risk of a temporarily invisible illness has always been there, and it is still there long after the family returns to their home and daily life. The GMA method is precisely used to spot signs of illness in a child before they “really” show, a skill reserved for the most specialized experts. This makes it difficult for some parents to be confident in reading and understanding their own child’s signals. Are a child’s particular movements, body language, sounds, or preferences normal? Normal for a premature infant? A sign of the child’s personality, or a sign of illness? “*I watch him closely, more than the other two. I don’t know why. Or maybe I do, it’s because I worry. I look at his hands and wonder if he’s moving them right. The physiotherapist says he’s on track, like others at that age. But I do watch him closely*”, a father explains. A mother supplies: “*Sometimes, when she gets toys, she gets these shakings. But the doctor said it’s a normal reaction, it shows that she is eager. And it’s good to know what things are, because we do wonder if things are normal, and she looks like she has Parkinsons, like, all shaky*”.

For some, this goes so far as to make them question their competence as caregivers. Understanding and interpreting their high-risk child’s signals has become the domain of medical specialists. These parents speak of frequently consulting services such as WellChild nurses and pediatricians for help to make judgment calls in ordinary situations, e.g., if a child spits up or cries more than usual, out of fear of not making optimal decisions or missing a subtle sign of illness. While some parents talk of trying to observe their child’s movements, looking for signs of CP, others emphatically state that they are not qualified to make such assessments and that they rely on the expert’s opinions to pass judgements over their child’s development and the presence or absence of signs of illness. Either way, judgements about the child’s movements, skills, and performances and the results of the GMA and other assessments wedge themselves into the parent–infant relation and are integrated into parents’ own knowledge of their child and their way of being.

Trust in the specialized competence and availability of the child’s follow-up team, whether local or hospital-based, is essential for these parents. As they rely on the results of assessments as a basis for their future, they also utilize support from people around the family to help cope. Some parents share stories of not being properly met and understood, sometimes having to air concerns multiple times to different nurses, doctors, and physiotherapists before meeting someone with sufficient expertise on high-risk infants to be able to confidently lay a concern to rest. Conversely, parents who experience being reassured, taken seriously, understood, and supported place a high level of confidence in the expert opinions of their follow-up team. Parents who see that professionals have gotten to know their child over time express an impression that they can see past momentary performances of symptoms and signs and include the child’s personality, developmental trajectory, and preferences in the totality of their professional assessments. With a close and trust-based relationship with the child’s follow-up team, parents feel increasingly comfortable with considering these assessments as true and as part of the foundation of how they themselves see and know their child. A parent elaborates on his experience with a follow-up team: “*We got more [information] when we got a physiotherapist who saw the child more regularly and could see what she was like. Same with the well child nurse and preschool, they understand her personality in it all. If she is annoyed and doesn’t want to do something one day they try another day and see all that she can manage*”.

When the prospects for health, fulfillment, and self-realization for a child are uncertain, the future demands on parenthood are also uncertain. To ensure that they are doing everything within their power to help their child and to secure their ability to meet their child’s potential future needs, parents use the results of early predictive examinations and their relationship with their child’s follow-up team to prepare. This is a safety net highly valued by parents. Yet, when integrating professionals’ objective assessments into their own understanding of their child, some also accept the implicit limitation on their own sufficiency. The way parents normally know and connect with their child does not seem enough when the child is considered high-risk. In order to be prepared and to ensure their ability to fulfill the parental role no matter what the future holds, parents accept this added layer of objectification, integrating the perspective of expert assessments into how they see their own child, just in case.

### 3.2. Indicating Variations in Experiences by Risk Status

Parents’ experiences with receiving different results of risk assessments give variations in the essential meaning structures of early predictions that are described under each meaning structure above. We find that the experiences parents share, such as beginning parenthood with trauma, the awareness of risk and of coping with uncertainty for months, form the essential meanings of early risk assessment, even before the GMA results are available to parents. Thus, the general meanings of “on a spectrum from death to insignificancies”, “living with uncertainty of what the parental role will entail”, and “seeing your child through the eyes of professionals, just in case” hold true regardless of the risk assessment result. What may vary depending on which result parents receive is where on the spectrum of severity parents eventually “land” their concerns, to what extent the burden of uncertainty is relieved or amplified, and ultimately, how parents can go about creating a sense of security for the future. When the result of the GMA indicates a low risk of CP, this helps reduce parents’ sense of alarm, but when the results do not reassure parents that their child is healthy, the opposite may occur.

## 4. Discussion

The aim of this study was to investigate the phenomenon of early predictive examination for CP as it appears to parents of high-risk infants when they do not yet know for certain whether their child has CP or not. The essential meaning structures we describe are as follows: 1: “on a spectrum from death to insignificancies”, 2: “living with uncertainty of what the parental role will entail”, and 3: “seeing your child through the eyes of strangers, just in case”. These must be understood against a background of the interplay between parenthood, the trauma of a threat to the new child, and the need for security. A traumatic birth or NICU experience generates a medical risk, which is the indication for predictive assessments and is therefore always a part of the experience and sense-making process for parents. The child’s life, well-being, and fulfillment and the aspects of parenthood are at stake with the results of early predictive assessments.

What some parents describe living through for weeks and months while predictive assessments take place constitutes an existential crisis [32]. During this time, the aspects and existence of parenthood are threatened by the risk posed to their child. This may explain why parents understand “early predictive examination for CP” broadly as anyassessment or examination which may unveil signs of future disability or illness. As they weigh their concern for their child’s health and future on a figurative scale from death to insignificancies in terms of how impactful the threat to their child may be, CP does not necessarily stand out as being more important in their minds than other conditions. All assessments, whether for CP or for other illnesses, inform parents about the severity of the threat to their child’s future. “On a spectrum from death to insignificancies” shows us that all adverse outcomes and all predictive assessments, for parents, ultimately amount to the same significant meaning: whether the child will be able to live, live happily, and live with fulfillment.

Living with awareness of the risk and managing the ensuing uncertainty is a state which lasts for months for parents of high-risk infants. We find that parents’ perceptions of their child’s risk and prospects influence their own sense of well-being, and that exhaustion, worrying, and loneliness are prominent for some. Strains on the mental well-being of parents of high-risk children are well documented in the existing literature, which has found anxiety, depression, PTSD, psychological distress, and parental stress impacting parent–child interactions to be more prevalent in parents of premature infants than in parents of term-born infants, and these conditions last for years after the children’s birth [33]. Challenges to the attachment between parents and high-risk infants is also well established, and interventions to reduce separation and promote parental mental health may alleviate some of these [34]. It is in this context that the results of predictive assessments are perceived. We find that parents who receive assessment results indicating a low risk of CP express relief and the ability to lay some of their worries to rest. This may help them to move forward and look to the future, even if they still have concerns for issues other than CP, which are deemed “minor”. Whether these positive sentiments in the early phase of the child’s development has a lasting impact on parents’ well-being and the parent–infant relationship warrants further studies. Furthermore, as shown in this study, it is noteworthy that the burden of uncertainty is amplified when parents do not obtain a clear answer from predictive assessments. A perception that “something is not quite as it should be, though there are no clear signs of CP” is difficult to navigate. This message does not lay parents’ concerns for the future to rest, nor does it help them to prepare as it does not inform them about what to prepare for. This suggests that paying attention to parents’ mental health and coping is especially needed when the results conveyed to parents do not clearly indicate a “high risk” or “low risk” of CP. This might not be needed to the same degree when early predictions conclude with a “low risk”.

The implication of “seeing one’s child through the eyes of strangers, just in case” should be considered when discussing attachment difficulties and challenges to the mental health of parents of high-risk children. This essential meaning constituent describes tension between parents’ need for security and the risk of objectification, contributing to a wedge in the parent–infant relationship. The phenomenological concept of the subjective and objective body is summarized by Danish psychologist Bo Jacobsen: on the one hand, we have a body (i.e., the body as object); on the other hand, we are our bodies (i.e., the body as subject). We have a body when we struggle with it or sense others judging its appearance. We are our body when we express ourselves through it or feel pain or well-being [32]. When facing illness, we lean towards the body as an object, in experiencing a body which “does not cooperate” or perform to the level we feel represent “who we really are”. How objectification, by way of assessing for signs of illness, plays out in the minds of the parents within the lived relationship between parent and infant is not well described from a phenomenological viewpoint. Our results indicate that highlighting the risk and importance of detecting signs of illness early on, as well as parents’ acceptance of outsiders’ judgements about their child’s signs through predictive assessments, in some instances, direct parents’ attention to the performative aspects of their child’s body and movements. “I am responsive and connect in the moment” and “I must detect subtle signs of abnormal development or illness” are qualitatively different ways of being sensitive to and “seeing” one’s child. It is interesting to consider whether a shifting balance between the subjective and objective in the parent–infant relationship, together with separation, exhaustion, and worrying, may play a causal role in challenges with attachment and the quality of parent–infant interactions [34,35]. Understanding this more fully requires more research aimed at the challenging act of weighing medical needs for security and predictive certainty against the impact on the parent–infant relation. Whether this shift in how parents observe their child evolves with time or changes depending on which risk assessment result parents have received is another question raised but not answered by our results.

Our study investigated parents’ experiences during months of uncertainty lasting from birth to after the GMA result was conveyed. Parents are aware that their child is de facto labeled as a high-risk infant at this time regardless of whether they will receive the interim clinical diagnosis of having a high risk of CP at some point. In line with previous research, our results confirm that parents appreciate the intention behind early predictive assessments being able to help the child and the family early on, that parents need to know that they are doing everything they can for their child, and that this induces a strong preference for early predictive assessments [18,19,22,36]. Furthermore, according to Brown et al. and Øberg et al., having an active role in predictive assessments and early intervention may be significant for the parents’ relationship with their infant, and building knowledge together with skilled professionals can reduce parents’ fear [37,38]. Our finding that early predictive assessments may simultaneously shift the balance between objective and subjective in the parent–infant relation is, however, a new perspective that has not previously been described. Still, we also find indications of such objectification in results like those from Morgan et al.’s concept of “New, Time-Consuming Challenges”, where emphasis on training and intentional interactions as a part of early intervention is described as “taking the fun out of it”, and in Cameron et al.’s idea of “Watching through a deficit lens”, where parents describe both themselves and others observing their child differently, looking for problems [22,36]. The full scope and significance of how early predictive assessments may impact the parent–infant relationship, in both the short and long term, is not known and warrants further studies.

The collective body of current literature on the parental experience with early prediction seem to support addressing attachment, parents’ mental well-being, and coping as early as possible regardless of whether the child will eventually be referred to a CP-specific early intervention program. The period of highest uncertainty for the parents begins immediately after birth, lasts for the following weeks and months, and gradually tapers when repeated predictive assessments show a low risk. This is also a formative period for the parent–infant relationship, which may be impacted by the life crisis parents go through. We further suggest that HCPs purposely look for and emphasize the infant’s individuality and strengths together with parents during clinical meetings both during the NICU stay and in outpatient and primary care settings. If parents receive uncertain results from their predictive assessments, we suggest following up by explicitly addressing parental coping, well-being, and attachment at this time, with further mental health counseling and attachment support being made available if needed.

This study uses a lifeworld approach to investigate the lived experiences of parents who mostly received low-risk and uncertain risk assessment results at a point in time when a definitive CP diagnosis is not made nor excluded. We describe a more complicated and nuanced view of early predictive assessments than previous studies investigating the experiences of parents of children with CP [18,20,21,22]. This is not surprising, and our results do not contradict the notion that early diagnosis and early intervention are important when a child is diagnosed with CP. Rather, this study should be understood as a supplement illuminating the phenomenon of early prediction for CP from a less described point of view: of the many parents who experience a life crisis which may amount to CP, it may amount to something less serious or may amount to nothing at all.

Strengths and limitations: The strengths of this study include a reflexive lifeworld approach, strategic sampling, an open interview strategy adapted to accommodate parents’ perspectives, and an empirical phenomenological analysis supplemented by a discussion which draws on the existing knowledge in the field of early diagnosis of CP and follow-up of high-risk infants. The second author’s area of expertise is phenomenological and lifeworld-led research and evidence-based qualitative research. The participants all live in Central Norway and have their experiences from hospitals and follow-up programs in this region, which limits the transferability of the findings to settings in parts of the world with differently organized health care systems, cultural norms, and social security systems. However, studies in the international literature exist which are in line with our essential constituents, which enhance analytical validity, suggesting that our findings have relevance in an international context. While some informants initially received a result of a “high risk of CP” and could speak to that experience, none had a “high risk of CP” after all GMAs. This skews our results towards the perspectives of parents who received uncertain results or a low-risk result after the GMA. Our participants had consented to another study on the feasibility of home filming for the GMA; thus, it is possible that they hold a favorable view of early prediction, the GMA method, or research in general, which may have skewed our results. The first author being a pediatrician is both a strength and a weakness; her experience provided a framework for understanding the data while also possibly inadvertently influencing what became talking points for participants.

## 5. Conclusions

Repeated early predictions take place over the course of months while parents live through a life crisis and process a traumatic experience. During this period, they feel uncertainty about the future and worry about potential illnesses, and the parent–infant relationship may be affected. Predictive assessments help reduce parents’ sense of alarm and helps them move on when the GMA result indicates a low risk of CP. A GMA that does not give a clear-cut answer does not help parents navigate their uncertainty, which is then prolonged and amplified.

## Figures and Tables

**Table 1 jcm-14-02740-t001:** Strategic selection criteria and informant characteristics at the time of the interview.

Selection Criteria	Informant Characteristic	Informants (n)
Age (years)	<24	1
	25–34	7
	>35	6
Nationality	Norwegian	14
	Other	0
Gender	Male	6
	Female	8
Education level	High school or less	9
	University	5
Single parent	Yes	1
	No	13
Siblings in family	Yes	5
	No	9
Child’s risk factor	Extreme prematurity (GA 23–28)	5
	Prematurity (GA 28–32)	3
	Acute illness or asphyxia	6
Results of early predictions	Low risk of CP (fidgety +/++)	9
	High risk of CP (fidgety −)	0
	Unlikely CP, not normal (fidgety +/−)	3
	Contradicting	2

## Data Availability

Transcripts of the in-depth interviews underlying the analysis presented in this study are not available due to ethical considerations of anonymity and participant integrity in qualitative research.

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
