# Peer review of "Doing Everything We Can to Help Our High-Risk Newborns: A Qualitative, Lifeworld-Led Study of What Early Risk Assessment for Cerebral Palsy Means to Parents"

_jcm, 2025, doi:10.3390/jcm14082740_

Round 1
Reviewer 1 Report
Comments and Suggestions for Authors
Firstly, would like to that the authors for their interesting and relevant work. Bellow, a few suggestions for your manuscript: 1) Firstly, would recommend adding the study design type to your title 2) Would recommend further elaborating on this study's hypothesis and questions on the last paragraph of the introduction. 3) In your methods, recommend explaining how did you arrive at this particular study protocol, how it was chosen and validated 4) Please include IRB number 5) Please include details about the timeline of the study, inclusion and exclusion criteria, and how did you arrive at the this "n" of participants 6) In your discussion would recommend further elaborating on the comparison of this studies to other studies published in the literature about this same subject 7) Please follow instructions for authors for this study, avoid underlined terms and semicolons on the beginning of each paragraph
Comments on the Quality of English LanguageNo issues detected
Reviewer 2 Report
Comments and Suggestions for Authors
The authors conducted qualitative study using phenomenological, reflective life-world approach with the aim to investigate the experience of parents of high-risk infants and describe the meaning “early predictive assessments for CP” has for them before they know if their child has CP. The topic is clinically important as it may inform counseling strategies for such parents in clinical practice. Suggestions:
1) Please use a reporting guideline for qualitative research (e.g. COREQ or SRQR) and provide corresponding checklist in the appendix. Please also mention and cite which reporting guideline was used in the first line of the methods section.
2) In the analysis section (between lines 157-173) please mention what software was used for coding and analysis (e.g. NVIVO).
3) The results section: please consider organizing key themes in a tabular format. A table of key findings will improve accessibility of results for the readers.
4) Were there any differences in experiences/key themes between low risk vs unlikely vs contradicting results from early prediction? I see this mentioned in some places but it is scattered in the results section. Please consider adding a separate section "3.2" indicating differences in experiences/key themes by risk status.
5) In the discussion section, please draw similarities and differences between your findings and findings from previous qualitative research investigating parents' views on early diagnosis for infants diagnosed with cerebral palsy (for example, see the following paper: Morgan C, Badawi N, Novak I. "A Different Ride": A Qualitative Interview Study of Parents' Experience with Early Diagnosis and Goals, Activity, Motor Enrichment (GAME) Intervention for Infants with Cerebral Palsy. J Clin Med. 2023 Jan 11;12(2):583. doi: 10.3390/jcm12020583. PMID: 36675512; PMCID: PMC9866599.)
In particular, consider drawing how counseling approaches would be adapted (for both- timing and content of counseling) based on joint findings from your study and previous study mentioned above.
Thank you.
Round 2
Reviewer 2 Report
Comments and Suggestions for Authors
Thank you for submitting a revised version; I have no additional comments
Author Response
Thank you for the positive response and thorough consideration of our manuscript.